# Expanding Phenotype of Schimke Immuno-Osseous Dysplasia: Congenital Anomalies of the Kidneys and of the Urinary Tract and Alteration of NK Cells

**DOI:** 10.3390/ijms21228604

**Published:** 2020-11-15

**Authors:** Cristina Bertulli, Antonio Marzollo, Margherita Doria, Silvia Di Cesare, Claudio La Scola, Francesca Mencarelli, Andrea Pasini, Maria Carmen Affinita, Enrico Vidal, Pamela Magini, Paola Dimartino, Riccardo Masetti, Laura Greco, Patrizia Palomba, Francesca Conti, Andrea Pession

**Affiliations:** 1Nephrology and Dialysis Unit, Department of Pediatrics, S. Orsola-Malpighi Hospital Scientific Institute for Research and Healthcare (IRCCS), 40138 Bologna, Italy; cristina.bertulli@aosp.bo.it (C.B.); Claudio.lascola@aosp.bo.it (C.L.S.); francesca.mencarelli@aosp.bo.it (F.M.); andrea.pasini@aosp.bo.it (A.P.); 2Pediatric Hematology, Oncology and Stem Cell Transplant Division, Padua University Hospital, 35128 Padua, Italy; antonio.marzollo@unipd.it (A.M.); draffinita@gmail.com (M.C.A.); 3Unit of Primary Immunodeficiency, Academic Department of Pediatrics (DPUO), Bambino Gesù Childrens’ Hospital-Scientific Institute for Research and Healthcare (IRCCS), 00165 Rome, Italy; doria@uniroma2.it (M.D.); di.cesare@med.uniroma2.it (S.D.C.); 4Dialysis and Transplantation Unit, Pediatric Nephrology, Department of Woman’s and Child’s Health, University Hospital of Padua, 35128 Padua, Italy; enrico.vidal@inwind.it; 5Medical Genetics Unit, Department of Medical and Surgical Science, S. Orsola-Malpighi Hospital_Scientific Institute for Research and Healthcare (IRCCS), 40138 Bologna, Italy; pamela.magini@aosp.bo.it; 6Medical Genetics Unit, DIMEC, University of Bologna, 40138 Bologna, Italy; paola.dimartino3@unibo.it; 7“Lalla Seràgnoli”, Hematology-Oncology Unit, Department of Pediatrics, University of Bologna, 40138 Bologna, Italy; riccardo.masetti5@unibo.it (R.M.); andrea.pession@unibo.it (A.P.); 8Pediatric Radiology Unit, Department of Diagnostic and Preventive Medicine, S. Orsola-Malpighi Hospital, 40138 Bologna, Italy; laura.greco@aosp.bo.it; 9Immunology Research Area—Unit of Diagnostic Immunology, Unit of B-cell Pathophysiology, Department of Laboratories, IRCCS Ospedale Pediatrico Bambino Gesù, 00165 Rome, Italy; patrizia.palomba@opbg.net; 10Pediatric Unit, Department of Woman, Child and Urologic Diseases, IRCCS Azienda Ospedaliero-Universitaria di Bologna, 40138 Bologna, Italy

**Keywords:** congenital, hereditary, neonatal diseases and abnormalities, consanguinity, DNA methylation, immune system diseases

## Abstract

Schimke immuno-osseous dysplasia (SIOD) is a rare multisystemic disorder with a variable clinical expressivity caused by biallelic variants in *SMARCAL1*. A phenotype–genotype correlation has been attempted and variable expressivity of biallelic *SMARCAL1* variants may be associated with environmental and genetic disturbances of gene expression. We describe two siblings born from consanguineous parents with a diagnosis of SIOD revealed by whole exome sequencing (WES). Results: A homozygous missense variant in the *SMARCAL1* gene (c.1682G>A; p.Arg561His) was identified in both patients. Despite carrying the same variant, the two patients showed substantial renal and immunological phenotypic differences. We describe features not previously associated with SIOD—both patients had congenital anomalies of the kidneys and of the urinary tract and one of them succumbed to a classical type congenital mesoblastic nephroma. We performed an extensive characterization of the immunophenotype showing combined immunodeficiency characterized by a profound lymphopenia, lack of thymic output, defective IL-7Rα expression, and disturbed B plasma cells differentiation and immunoglobulin production in addition to an altered NK-cell phenotype and function. Conclusions: Overall, our results contribute to extending the phenotypic spectrum of features associated with *SMARCAL1* mutations and to better characterizing the underlying immunologic disorder with critical implications for therapeutic and management strategies.

## 1. Introduction

Schimke immuno-osseous dysplasia (SIOD) (OMIM 242900) is a rare autosomal recessive multisystemic disorder characterized by spondyloepiphyseal dysplasia, growth retardation, renal impairment, and T-cell dysfunction [1,2,3]. Cerebrovascular events, premature atherosclerosis, hypothyroidism, ectodermal abnormalities such as altered tooth development have been described. Patients with SIOD are at increased risk of developing cancer, mainly osteosarcoma, undifferentiated carcinoma, and non-Hodgkin lymphoma [4,5,6,7,8,9]. Early death is common and is mainly related to opportunistic infection or end-stage renal disease.

Schimke immuno-osseous dysplasia is caused by biallelic variants in *SMARCAL1*, encoding the SWI/SNF-related, matrix-associated, actin-dependent regulator of chromatin subfamily A-like protein-1 (SMARCAL-1) from the SWI2/SNF2 family of ATP-dependent chromatin remodeling proteins [10,11,12]. Disease severity inversely correlates with residual SMARCAL1 activity—mild and severe forms of SIOD are associated with missense and predicted loss-of-function (LOF) variants (nonsense, frameshift, or splicing), respectively [13]. SMARCAL1 plays an important role in DNA stabilization and its deficiency leads to the impairment of cellular function due to the accumulation of DNA damage, resulting in a progressive systemic disease. It is still not clear how functional impairment of SMARCAL1 causes such a specific SIOD phenotype represented by alterations in the bones, kidney, and immune system, however it has been shown that proteins encoded by *SMARCAL1* orthologs buffer fluctuations in gene expression and that alterations in gene expression contribute to SIOD manifestations [14,15,16]. Cell-mediated immune defects have been partially explained by the lack of IL7Rα expression and IL-7 responsiveness due to hypermethylation of the IL7R promoter in SIOD T cells thus representing a hallmark of T-cell immunodeficiency in SIOD [17].

We describe two brothers born from consanguineous Moroccan parents, with different clinical courses of SIOD, due to the same homozygous c.1682G>A (p.Arg561His) variant in *SMARCAL1*.

## 2. Results

### 2.1. Clinical Report

#### 2.1.1. Patient 1 (P1)

The index case was born from consanguineous Moroccan parents (first cousins) at 32 weeks of gestation with a weight and length below the 3rd centile and with a prenatal diagnosis of multiple malformations, including a single umbilical artery, duplication of the choroid plexuses with mild lateral ventriculomegaly, a hypoplastic corpus callosum and cerebellar vermis, cardiovascular anomalies, abdominal calcifications, hyperechogenic bowel, and rocker-bottom feet. After birth liver calcifications, a right ectopic kidney, an atrial septal defect type II, and asymmetric ventriculomegaly were found. Persistent leukopenia and lymphopenia led to the diagnosis of combined immunodeficiency (CID) and the introduction of cotrimoxazole prophylaxis. After birth a right-sided hemiparesis was noted. Cerebral MRI performed at 4 months showed a malacic area located at the lentiform nucleus and external left capsule, with dilatation of the subarachnoid spaces of the insula and left lateral ventricle, consistent with a previous cerebral ischemic event. The exam was repeated 2 years later showing the same alterations (Figure 1).

During infancy he suffered from several upper respiratory tract infections and one episode of intestinal amebiasis caused by *Entamoeba histolytica*. Severe episodes of migraine-like headaches were reported and psychomotor and growth retardation was observed. At the age of four, he attended kindergarten with school support, he had not attained sphincter control and language skills were almost absent; on clinical examination, he presented growth retardation (weight and height below the 3rd centile) and some dysmorphic features—microcephaly, sparse eyebrows, epicanthus, telecanthus, wide nasal root, blue sclerae, prominent metopic, and protruding ears with a thin helix. At the age of five, he presented with steroid-resistant nephrotic syndrome which progressed to chronic kidney disease and was managed with conservative treatment. Detection of hypogammaglobulinemia led to the initiation of intravenous immunoglobulin (IVIg) replacement therapy. The clinical features of the patient associated with the typical skeletal abnormalities (left convex thoracic scoliosis, abnormal femoral heads, and dismorphic vertebrae), led to the suspicion of Schimke immuno-osseous dysplasia (SIOD) (Table 1).

#### 2.1.2. Patient 2 (P2)

The younger brother, carrying the same homozygous variant in *SMARCAL1*, was born at 31 weeks of gestation, with a prenatal diagnosis of intrauterine growth restriction, multicystic left kidney, and a neoformation on the right kidney. Furthermore, hyperpigmented macules on the right arm were reported at birth and oral levothyroxine was started for congenital hypothyroidism. Leukopenia and persistent lymphopenia were detected with an immunological phenotype consistent with a diagnosis of combined immunodeficiency (CID). Moreover, ecochardiography revealed an interatrial defect with pulmonary stenosis. An abdominal MRI scan confirmed the presence of a multicystic left kidney (MCDK) and a mass on the right kidney which was hard to distinguish from the overall kidney parenchyma (Figure 2).

Open surgical biopsy of the lesion was consistent with classical-type congenital mesoblastic nephroma (CMN). Molecular detection of the *ETV6*-*NTRK3* gene fusion was negative, thus excluding a possible congenital cause of the tumors. The right renal mass associated with the left MCDK caused progressive chronic kidney disease (maximum sCr 2.09 mg/dL) with the presence of residual diuresis. Given the patient’s low body weight, the residual renal function maintained only on a small portion of functional right kidney (not infiltrated by the tumor) and the benign nature of the CMN, warranted a ‘wait-and-see’ approach. Five months later, the nephroma had increased in size to 11 cm causing respiratory distress and reduced oral tolerance with failure to thrive. Chemotherapy with vincristine was started (0.05 mg/kg/dose weekly for 6 weeks), with the aim of reducing the CMN in order to perform nephron-sparing surgery. The treatment successfully reduced the tumor volume to 4.5 cm, but unacceptable toxicity (prolonged grade 4 neutropenia and repeated sepsis), associated with severe psychomotor delay and growth retardation, led to treatment interruption. Therefore, an attempt of surgical removal of the mass was attempted which resulted in a partial debulking only and the patient developed end-stage kidney disease requiring kidney replacement therapy. Unfortunately, the tumor progressed a month later. Thus, vincristine treatment was resumed and 9 weekly doses were administered during the period in which the patient was being treated with IVIg, antifungal mold-active and antiviral prophylaxis. The treatment was well-tolerated, however the tumor continued to expand and infiltrate contiguous structures. Palliative treatment was administered until death at 15 months of age.

### 2.2. WES Findings

WES analysis was performed on P1. A mean coverage of 145X was obtained and 98.3% of bases were covered >20X. Two homozygous variants survived the prioritization process (Table 2), but only the p.Arg561His in *SMARCAL1* emerged as the most prominent candidate, with a CADD score of 32 and affecting a disease gene causing a clinical condition similar to that presented by the two siblings, Schimke immunoosseous dysplasia (MIM 242900). The homozygous *SMARCAL1* variant was also present in P2 and both parents were heterozygous carriers (Figure 3).

This missense change, mapping in a run of homozygosity (ROH) of about 5 Mb, involved a highly-conserved residue; it was classified as likely pathogenic by Varsome (https://varsome.com/), according to the American College of Medical Genetics (ACMG) guidelines [18], and it had been previously reported in two patients with clinically-diagnosed Schimke immuno-osseous dysplasia (SIOD), in a compound heterozygous state with another variant (c.3G>A, p.Met1Ile) in one case [19] and in a homozygous state in the other one [20,21] (Table 1).

### 2.3. P1 and P2 Showed a Combined Immunodeficiency Phenotype

Immunophenotyping of the peripheral blood mononuclear cells (PBMCs) revealed that both patients had significant T-cell lymphopenia characterized by low naïve CD4+ T-cell (CD4+CD45RA+) and CD8+ T-cell (CD8+CD45RA+) counts (Table 3). The frequencies of B cells (CD19+) and NK cells (CD3-CD56+CD16+) were close to those of age-matched controls. P1 and P2 B phenotype showed normal subset distribution compared to age-matched values except for an increased percentage of transitional B cells (CD38++IgM++). After stimulation with CpG oligonucleotide in vitro, B cell proliferation was reduced (Figure 4A) and differentiation into plasmablasts was abolished (Figure 4B). Accordingly, no immunoglobulin production was found in the supernatant (Figure 4C).

The immunoglobulin levels in both patients’ sera showed a reduction of IgG with normal levels of IgA and IgM. IL-7Rα (CD127) expression on P1 T lymphocytes was severely reduced on total CD3+ and CD4+ cells and almost absent on CD8+ T cells, while the mother’s CD4+ and CD8+ T lymphocytes showed only a slightly decreased CD127 expression (Figure 5).

The NK-cell compartment was further analyzed in P1 showing a strong expansion of the CD56^bright^ NK-cell subset [45.2% vs. 13.6% (12.9, 15.6), median % (interquartile range, IQR) of Healthy Donors (HDs)], at the expense of CD56^dim^ cells (46.0% vs. median 65.8% (60.0, 69.2) of HDs) (Figure 6A,B). The majority of CD56^bright^ cells in the SIOD patient did not express CD16 and had an immature phenotype, with an overall expansion of CD56^bright^CD16^−^NKG2A^+^NKG2C^−^CD57^−^ cells (29.7% vs. 13.7% (12.8, 16.2) of HDs; Figure 6C), which has been previously described in children with X-linked severe combined immunodeficiency (X-SCID) [25]. A further analysis of the NK-cell phenotype showed that the patient presented a normal pattern of expression of the NKG2A and KIR inhibitory receptors and DNAM-1 and NKG2D activating receptors (Figure 6D shows comparison with a representative HD). On the other hand, the intracellular levels of perforin, which is highly expressed by virtually all CD56^dim^ cells in HDs, were reduced by 3-fold in the patient’s CD56^dim^ cells (Figure 6E). Moreover, we found that IL-7Rα (CD127) expression, which is typically restricted to the CD56^bright^ subset within NK cells, was present on the patient’s CD56^bright^ cells albeit reduced by 4-fold (Figure 6E). Next, we analyzed the cytotoxicity of the patient’s NK cells against K562 cell targets by measuring the expression of the CD107a degranulation marker. Figure 6F shows that the cytotoxic activity of the patient’s NK cells was strongly reduced when compared with cells of HDs.

## 3. Discussion

We describe two brothers with SIOD carrying the same missense homozygous variant in *SMARCAL1*, with different clinical manifestations and degrees of severity. P1 presented with growth retardation, stroke, skeletal anomalies, developmental delay (especially language), cardiac and renal malformation and immunodeficiency since birth and he developed severe nephrotic syndrome at 5 years of age. Despite the presence of a combined immunodeficiency, he did not experience any severe infections. P2 presented with MCDK, a clinically-aggressive classical type CMN, growth retardation, hypothyroidism, and combined immunodeficiency. Even though an unclear genotype–phenotype correlation was previously supposed, suggesting milder effects for missense variants and severe consequences for LOF molecular events, the observation of intra- and inter-familial clinical variability revealed that disease severity or outcome cannot be predicted by *SMARCAL1* genotype [2,26,27]. Nonetheless, the available literature suggests that missense variants seem to allow for some residual *SMARCAL1* function, causing a mild SIOD phenotype, while nonsense, frameshift or splicing variants consistent with a loss-of-function (LOF) mechanism cause the more severe form [2,26]. Both siblings developed a severe SIOD phenotype (onset of disease symptoms in the first year of life, growth failure, nephropathy, abnormal thyroid function, severe infections, and cerebrovascular symptoms). In addition to the well-known SIOD signs and symptoms, they had atypical clinical features, including congenital right ectopic kidney in P1 and MCDK affecting the left kidney associated with nephroma in P2 as well as an atrial septal defect and cerebral anomalies (ventriculomegaly and hypoplasia of the corpus callosum) in P1 and interatrial defect and pulmonary stenosis in P2. Within the spectrum of congenital anomalies of the kidneys and of the urinary tract (CAKUT), ectopic kidney and MCDK have never been reported as a manifestation of SIOD to date. The homozygous p.Arg561His variant has already been described in a patient with early-onset SIOD, who developed non-Hodgkin lymphoma [28]. In this previous study, the SIOD patient, despite immunodeficiency, did not experience recurrent infections or ischemic events, supporting a milder phenotype compared to the siblings presented herein. Moreover, no congenital kidney, heart, or brain anomalies were observed in that patient. A similar mild form of SIOD was described in a patient with a different homozygous change at the same residue (p.Arg561Cys). Severe clinical manifestations were reported in a boy carrying the p.Arg561His variant in the compound heterozygous state with the c.3G>A (p.Met1Ile) change, causing a start loss and generating a predicted truncated protein missing the nuclear location signal [20]. Clinical features of SIOD patients with variants involving the Arg561 residue are summarized in Table 1. The Arg561 residue lies within the helicase ATP-binding domain (amino acids 445–600, Uniprot database, https://www.uniprot.org/) and its substitution with a histidine could interfere with DNA binding and/or ATP hydrolyzation, potentially leading to a LOF effect. All these findings indicate that variants of Arg561 residue are likely responsible for SIOD with variable severity, depending on the resulting amino acid, on the type of alteration on the other allele, and on the individual genomic background. We searched for additional WES pathogenic variants contributing to the complex phenotype of P1, especially to extra-SIOD features. No homozygous variants in disease genes, except the *SMARCAL1* variant, were known or predicted to be pathogenic or were previously associated with congenital kidney and heart defects. The evaluation of heterozygous variants excluded the co-occurrence of dominant or recessive (through compound heterozygosity) genetic diseases modifying the typical SIOD phenotype. Unfortunately, the impossibility of extending WES analysis to P2 due to technical issues, prevented us from investigating additional genetic variants underlying his atypical manifestations, especially the congenital mesoblastic nephroma. CMN is a rare tumor in children representing approximately 5% of all pediatric renal tumors. Three different types of MN are distinguished histologically–classical, cellular, and mixed. Treatment consists of nephrectomy or nephron-sparing surgery at birth. Preoperative chemotherapy with actinomycin and vincristin can be administered [29]. Biological target therapy can be considered in the presence of *ETV6-NTRK3* gene fusions [30]. Even though malignancies are reported in SIOD patients, nephroma has never been observed. In a review of 71 patients, two developed Epstein–Barr virus (EBV)-positive, non-Hodgkin lymphoma, one EBV-negative non-Hodgkin lymphoma, and one osteosarcoma [14]; a case of undifferentiated carcinoma has also been reported [5]. Literature regarding the relationship between SMARCAL1 and neoplasia is poor. EBV-driven lymphomas can be attributed to immunodeficiency, but other malignancies reported in SIOD patients are less clearly associated with immune system disorders. SIOD is not currently considered a cancer predisposition syndrome and the incidence of malignancy in these patients is not well described, due to the limited number of cases and short life expectancy. However, members of the SWI/SNF2 family of ATPases are frequently mutated in human cancer, behaving as tumor suppressors, and the function of SMARCAL1 in DNA transcription, replication, and repair could play a role in the development of malignancy if the gene is mutated [31].

Regardless of a possible pathogenetic link, in the study, we describe the first classical-type mesoblastic nephroma in a patient with SIOD (P2) with implications for cancer treatment. In our patient, the tumor showed an unusually aggressive course with transient response to chemotherapy and failure of surgical treatment. It is well known that patients with SIOD are hypersensitive to DNA-damaging agents, so chemotherapy should be administered with caution, and even in the case of a transplant (kidney or bone marrow) a reduced-intensity conditioning must be adopted [14]. Both patients showed a combined immunodeficiency as proven by a profound lymphopenia, a lack of thymic output, and defective IL7Rα expression on lymphocytes. These results are consistent with data already reported in the literature [17]. We also show that B cells have a reduced ability to proliferate and differentiate into plasmablasts in vitro resulting in a lack of immunoglobulin in the supernatant (Figure 4). As IL7 is not necessary for the development of B cells and their response to CpG [32] impaired epigenetic remodeling may explain the altered B cell function [33]. Furthermore, we have characterized the NK-cell compartment of a SIOD patient for the first time, highlighting phenotypic and functional abnormalities. In particular, P1 presented a relative increment of the CD56^bright^ cell subset at the expense of CD56^dim^ cells. In addition, the patient’s NK cells displayed a normal phenotype with the exception of a strong down-modulation of IL-7Rα on CD56^bright^ cells and intracellular perforin in CD56^dim^ cells. In a healthy condition, the highly cytotoxic CD56^dim^ cells represent the vast majority of peripheral NK cells, whereas CD56^bright^ consist in a small percentage of NK cells that are considered to be immature precursors of CD56^dim^ cells, but yet can release large amounts of cytokines and exert immunoregulatory functions (e.g., killing of activated immune cells) [34]. The significance of the increased CD56^bright^ cell frequency that is found in various pathologic settings such as chronic viral infection, cancer, and autoimmunity, is not clear, as yet, and a potential immunosuppressive role has been suggested [26,35]. One possible leading mechanism for accumulation of CD56^bright^ cells is a decreased rate differentiation towards mature CD56^dim^ cells, hence it is possible that the SMARCAL-1 mutation in P1 had a negative impact on NK cell development, although not as dramatic as for T cells. Of note, IL-7Rα expression in our patient was nearly absent on T cells, a previously-reported hallmark of SIOD patients that may restrict T-cell development [17], while it was present, albeit reduced, on CD56^bright^ NK cells. This suggests that the mechanisms controlling IL-7Rα expression are distinct in T and NK cells and that residual IL-7 signalling in the patient’s CD56^bright^ cells is sufficient for their homeostasis. Importantly, IL-7Rα down-modulation on CD56^bright^ NK cells occurring in chronic Hepatitis C Virus (HCV) and HIV infection has been shown to impair IL-7-dependent NK-cell activation and effector functions [36], a phenomenon that deserves to be investigated in SIOD patients as well. Finally, we found an overall impairment of NK cell cytotoxicity, possibly due to the reduced frequency and low perforin content of CD56^dim^ cells, which contribute to the immunological dysfunction and increased risk of developing tumors or severe infections in the SIOD patient.

Taken together these results led us to consider these patients as leaky-SCID and thus evaluable for hematopoietic stem cell transplantation as a therapeutic option to treat the immunodeficiency. In the literature few cases have been reported with poor outcomes for this procedure. Only one affected individual has been successfully treated with bone marrow transplantation (BMT) [14,37] thus this procedure should be considered for the severe phenotype only. Factors described as a possible cause for a poor outcome were a debilitated state of the patient and the potential role of SMARCAL1 in cell hypersensitivity to genotoxic agents. However, the low number of bone marrow transplants and the heterogeneity of the patients’ genotypes and phenotype as well as the diversity of the procedures do not shed light on any specific predictors of outcome. Our knowledge about SIOD is still limited and, considering the rarity of the disease, multicentric studies are necessary in order to identify the best management of these patients.

## 4. Materials and Methods

### 4.1. Subjects and Families/Collection of Samples and Informed Consent 

The parents of the two siblings with SIOD gave written informed consent for the clinical evaluations and genetic analyses, in accordance with the ethical standards of the institutional research committee and with the 1964 Helsinki declaration and its later amendments or comparable ethical standards. The clinical data for the patients were obtained from questionnaires completed by the attending physician as well as from medical records. Ethics Committee approval was obtained along with written informed consent for data collection (protocol n. 138/2017/U/Tess approved date: 12 December 2017).

### 4.2. Whole Exome Sequencing/Sanger Sequencing

Genomic DNA purified from whole peripheral blood samples of P1 was enriched for whole exome sequences through the Roche SeqCap EZ MedExome Kit and sequenced as 100 bp paired-end reads on the Illumina NextSeq 500 system. Quality check for the generated reads was performed with FastQC (http://www.bioinformatics.babraham.ac.uk/ publications.html). Reads were aligned with Burrows-Wheeler Aligner (BWA) to the University of California Santa Cruz (UCSC) reference genome, hg19. Local realignment and base quality score recalibration was performed with Genome Analysis ToolKit (GATK) and duplicate removal with PicardTools (http://picartools.sourceforge.net). SAMtools and GATK were used to collect alignment statistics. Variants passing quality filters were annotated against Ensembl (http://www.ensembl.org/). Since the pedigree suggested a likely recessive disease (two affected siblings born from consanguineous healthy parents), we focused on biallelic, especially homozygous variants. Runs of homozygosity (ROHs) were detected from WES data through the H3M2 algorithm [38].

Novel or very rare homozygous variants (MAF < 0.001) within ROHs > 1.5 Mb were considered for further analyses when they caused predicted loss-of-function alterations or when PhyloP and pathogenicity prediction scores indicated conserved nucleotides and probably damaging affected amino acids for missense changes. The same filtering parameters (frequency, conservation, and functional-prediction scores) were used to identify candidate heterozygous variants potentially contributing to atypical clinical manifestations through a dominant or a recessive (compound heterozygosity) mechanism.

The quality and amount of the available DNA sample of P2 were not adequate for WES analysis.

Sanger sequencing was performed to validate filtered variants and to verify segregation in the healthy parents and affected brother (P2).

### 4.3. Flow Cytometry Analysis

#### 4.3.1. Immunophenotype and IL7Ra Membrane Expression

After red blood cell lysis with ammonium chloride of peripheral blood samples, lymphocytes were surface stained for T and B cell analysis. The following previously-titrated monoclonal antibodies were employed to surface stain the lymphocytes: CD3 PerCP (clone BW264/56, Miltenyi Biotec, Bergisch Gladbach, Germany), CD4 APC (clone OKT4, Becton Dickinson, Franklin Lakes, NJ, USA), CD8 PE-Cy7 (clone RPA-T8, Becton Dickinson, USA), TCR alpha-beta APC (clone T10B9, Becton Dickinson), TCR gamma-delta FITC (11F3, Miltenyi Biotec, DE), CD45RA APC-H7 (clone T6D11, Miltenyi Biotec, DE), CCR7 PE (clone 3D12, Ebioscience, San Diego, CA, USA), CD127 PE-CY7 (clone eBioRDR5, eBiosciences), CD16 PE (clone 3G8), CD56 PE (clone NCAM16.2), CD19 PE-CY7 (clone SJ25C1, Becton Dickinson), CD27 FITC (clone M-T271, Becton Dickinson). 

NK cell studies were performed on cryopreserved lymphocytes previously isolated through Ficoll density gradient and stained with NKG2A(CD159a) FITC (clone REA110, Miltenyi Biotec), NKG2C (CD159c) PE (clone REA205, Miltenyi Biotec); NKG2D(CD314) PE (clone 1D11, eBioscience), CD3 APC (clone UCHT1, eBioscience), CD16/APC-eFluor780 (clone CB16, eBioscience), CD57/PECy7 (clone TB01, eBioscience); CD3/AlexaFluor700 (clone UCHT1, Becton Dickinson), CD56/PerCpCy5.5 (clone B159, Becton Dickinson), CD16/BV510 (clone 3G8, Becton Dickinson), Perforin/BV421 (clone delta g9, Becton Dickinson), conjugated mouse IgG for isotype control staining (BD Pharmingen); CD56/PerCp (MEM-188) from Thermo Fisher Scientific (Waltham, MA, USA); CD107a/FITC (H4A3), DNAM-1(CD226)/FITC (11A8), NKp46/PE-Cy7 (9E2), KIR2DL1/S1/S3/S5/APC (HP-MA4), and KIR2DL2/L3/S2/APC (DX27) from Biolegend (San Diego, CA, USA). Cells were incubated with the appropriate antibody cocktail for 30 min at 4 °C, washed with PBS and suspended in PBS. For intracellular perforin staining, cells were reacted with FOXP3 Fix/Perm Buffer Set (Biolegend, USA) as recommended by the manufacturer. At least 50,000 events in the lymphocyte live gate were acquired on a FACSCANTO II (BD Biosciences, San Diego, CA, USA) or Cytoflex (Beckman Coulter, Brea, CA, USA) and analyzed with FlowJo (Tree Star Inc, version 9.3.2, Ashland, OR, USA) or Kaluza (Beckman Coulter, Brea, CA, USA) software.

#### 4.3.2. NK-cell Degranulation Assay

A flow cytometry-based cytotoxicity assay was performed using the patient’s PBMCs as effectors (E) and K562 cells as targets (T) at an E:T ratio of 10:1 and measuring the frequency of CD107a^+^ cells within gated NK cells, as previously described [19].

### 4.4. Cell Preparation and B Cell Proliferation Assay

PBMCs were isolated by Ficoll PaqueTM Plus (Amersham Pharmacia Biotech) density-gradient centrifugation. Lymphocytes were washed in PBS (1×) and before stimulation, peripheral blood mononuclear cells were labelled with Carboxyfluorescein succinimidyl ester (CFSE, Invitrogen) at a final concentration of 0.1 µg/mL (ThermoFisher Scientific) and cultured at 5 × 10^5^ cells per well in 96-well plates in complete Roswell Park Memorial Institute medium (RPMI) 1640 (Euroclone) supplemented with 10% FBS (Hyclone Laboratories, Logan, UT, USA) in the presence or absence of 2.5 µg/mL of CpG oligodeoxynucleotide (ODN 2006, Hycult Biotechnology, Uden, The Netherlands). Cell proliferation was measured on day 7 by flow cytometry.

#### 4.4.1. Flow Cytometric Analysis

To evaluate the proliferation and differentiation of B cells after stimulation with CpG, cells were stained with the appropriate combination of fluorochrome-conjugated Abs to identify B cell subsets: CD19 BB700 (clone SJ25C1 Becton Dickinson), CD27 PE (clone M-T271 Becton Dickinson), CD38 BV421 (clone HIT2, Becton Dickinson), and IgM Alexafluor647 (conjugated Affinipure F(ab’)2, Jackson ImmunoResearch Laboratories, West Grove, PA, USA). Dead cells were excluded from analysis by side/forward scatter gating. All analyses were performed on a LSRFortessaX-20 (Becton Dickinson, San Jose, CA, USA) interfaced to a FACSDiva software (BD Biosciences, San Jose, CA, USA). Fifty thousand gated events on living cells were analyzed, whenever possible, for each sample.

#### 4.4.2. ELISA Immunoassay for IgM, IgA, and IgG

ELISA immunoassay was employed to quantify plasma Igs and secreted Igs after stimulation with CpG. 96-well plates (Corning) were coated overnight with purified anti-human IgA, IgG, and IgM (Jackson ImmunoResearch Laboratories, PA, USA). Plates were washed with PBS/0.1% Tween and blocked with PBS/1% gelatin. Subsequently, two incubation steps for 1 h at 37 °C, first with culture supernatants and second with peroxidase-conjugated goat anti-human IgA, IgG, or IgM Abs (Jackons ImmunoResearch Laboratories, PA, USA) were performed. The chromogene substrate employed to developed the assay was *o*-phenylen-diamine solution (Sigma-Aldrich, St. Louis, MO, USA). Measurement of the absorbance at 450 nm and calculation of Ig concentrations by interpolation from the standard curve were performed lastly.

## Figures and Tables

**Figure 1 ijms-21-08604-f001:**
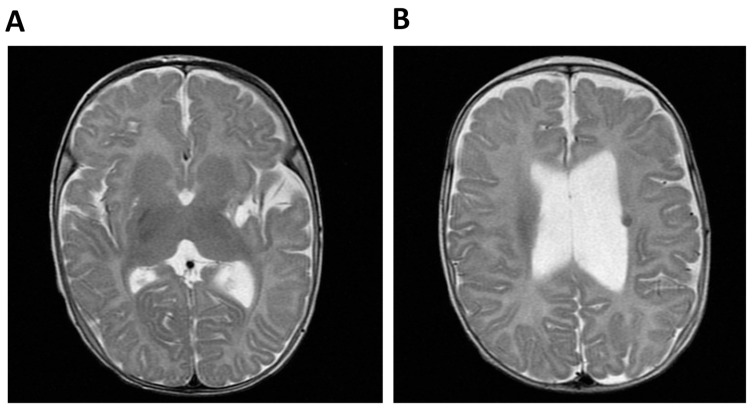
Cerebral MRI of Patient 1 (P1, index case). (**A**) Malacic area located at the lentiform nucleus and external left capsule and dilatation of the subaracnoid spaces of the insula and (**B**) Dilatation of the left lateral ventricle and T2-weighted hypointense lesion compatible with hemoglobin degradation.

**Figure 2 ijms-21-08604-f002:**
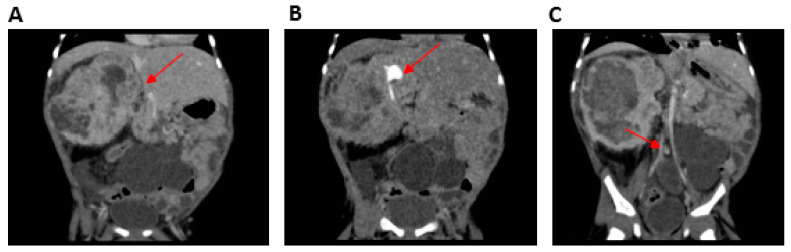
Computerized tomography (CT) urogramof Patient 2 (P2) (performed after vincristine therapy and before surgery). (**A**) Mesoblastic nephroma located in the right kidney (arrow), (**B**) contrast-enhanced portion of the functioning right kidney located in the superior pole of the nephroma (arrow) without a cleavage plane, and (**C**) multicystic left kidney (arrow).

**Figure 3 ijms-21-08604-f003:**
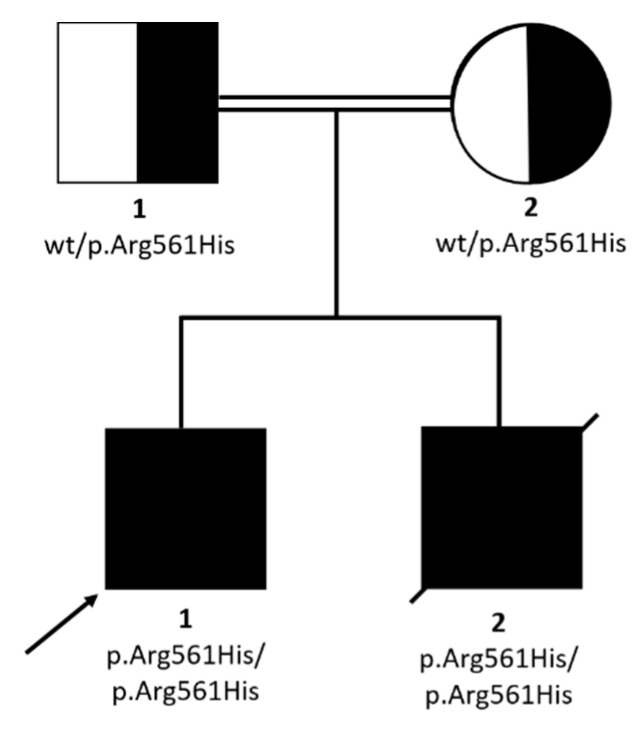
Pedigree and family segregation of *SMARCAL1* variant (c.1682G>A; p.Arg561His).

**Figure 4 ijms-21-08604-f004:**
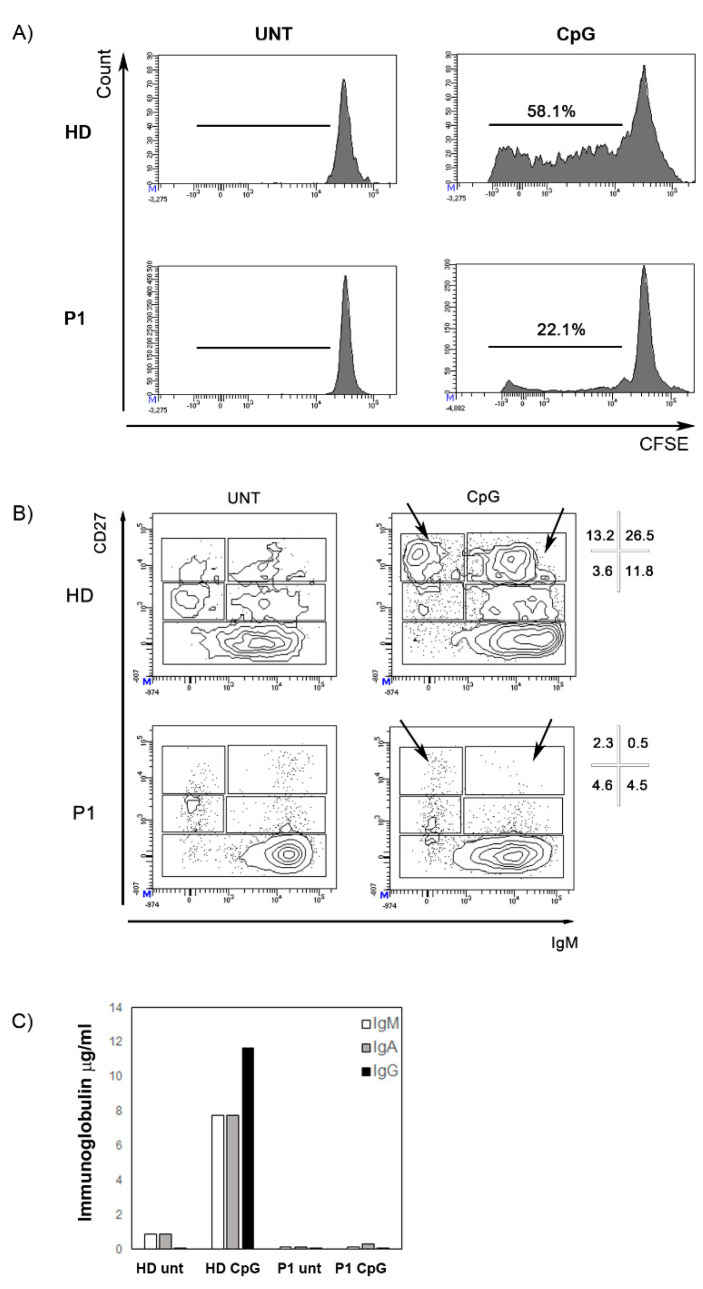
Stimulation with CpG. (**A**) The histograms represent B cell proliferation in a healthy control and P1. B cells were loaded with CFSE dye. The intensity of fluorescence decreased with cell division. The numbers indicate the percentage of B cells that had proliferated. (**B**) CD27 and IgM expression is determined in P1 and in a healthy donor with or without the addition of CpG (untreated, UNT). Different B cell populations can be recognized—CD27- mature/naive B cells, IgM+CD27+ memory B cells, IgM-CD27+ switched-memory B cells, and CD27bright plasmablasts (either IgM+ or IgM−, arrows). Percentages of these subpopulations in the CD19+ B cells gate are indicated. Black arrows indicate plasmablasts. (**C**) IgM (white columns), IgA (grey columns), and IgG (black) concentration (µg/mL) in the supernatants after 7 days of stimulation with CpG in a healthy control and in P1.

**Figure 5 ijms-21-08604-f005:**
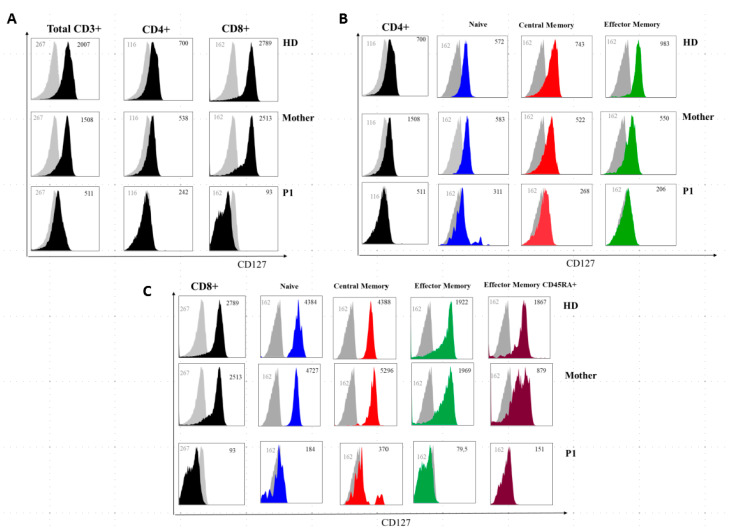
IL7Rα (CD127) surface expression on T-cell subsets from a healthy donor, P1, and the patient’s mother. (**A**) Grey histogram plots represent signals from isotype control staining. Note that total CD3+, total CD4+, and total CD8+ uniformly express reduced IL7Rα in P1 and are slightly reduced in the mother. (**B**,**C**) IL7Rα (CD127) surface expression of T-cell subsets of a healthy donor, the mother, and P1. Grey histogram plots represent signals from isotype control staining. Note that total CD4+, naive CD4+ cells (CD4+CD27+CD45RA+), central memory CD4+ cells (CD4+CD27+CD45RA and effector memory (CD4+CD45RA-CD27-) uniformly express reduced IL7Rα in P1 and are slightly reduced in the mother. (**C**) Note that total CD8+, naive CD8+ cells (CD8+CCR7+CD45RA+), central memory CD8+ cells (CD8+-CCR7+CD45RA−), effector memory (CD8+-CCR7-CD45RA−) and terminal effector memory CD45RA+ (CD8+-CCR7-CD45RA+) uniformly express severely-reduced IL7Rα in P1 and are slightly reduced in the mother. Grey numbers represent mean fluorescence intensities (MFI) of the isotype control staining and black numbers represent mean fluorescence intensities of the specific CD127 staining on the different T cell subsets.

**Figure 6 ijms-21-08604-f006:**
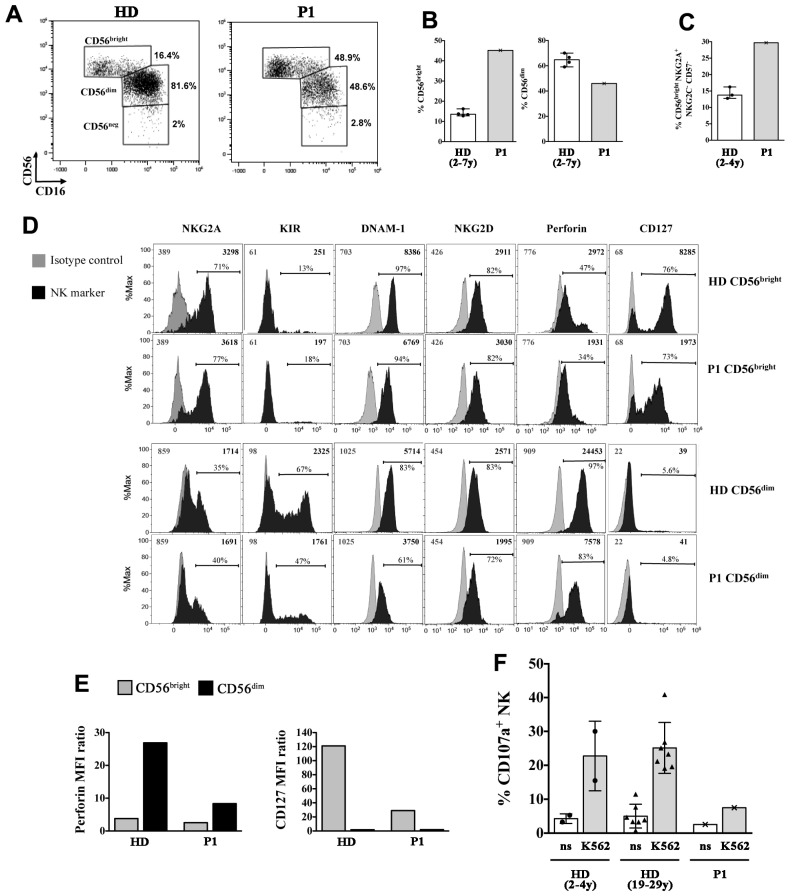
Phenotypic and functional alterations of NK cells in the P1 SIOD patient. (**A**–**C**) peripheral blood mononuclear cells (PBMCs) of P1 and of age-matched healthy donors (HDs) were analyzed by flow cytometry to measure the frequency of NK-cell subsets, the phenotype of gated CD56bright and CD56dim NK cells (**D**,**E**), and the NK-cell cytotoxicity (**F**). (**A**) The gating strategy used to identify CD56bright, CD56dim, and CD56neg NK cell subsets is shown for a representative HD and the SIOD patient, (**B**) Median and range of CD56^bright^ and CD56^dim^ cell frequencies among NK cells of the patient and four HDs is shown; (**C**) Median and range of CD56^bright^ cell percentage with a CD16-NKG2A+ NKG2C- CD57- immature phenotype among NK cells of P1 and three HDs is depicted, (**D**) Filled black histograms show the cell surface expression of NKG2A, KIR (KIRmix: KIR2DL1/S1/S3/S5 and KIR2DL2/L3/S2), DNAM-1, NKG2D, and CD127 as well as intracellular perforin expression of gated CD56^bright^ and CD56^dim^ NK cells of P1 and a representative HD. Filled gray histograms represents staining with isotype control IgG. The mean fluorescence intensity (MFI) values for both NK marker labeling (black) and control IgG (gray), as well as the percentage of positive cells are indicated, (**E**) The values of perforin MFI (left) and CD127 MFI (right) divided by their control IgG MFI values (ratio) measured on CD56^bright^ and CD56^dim^ cells as shown in (**D**) are reported, (**F**) Bar plots represent pattern of CD107a expression measured by flow cytometry on gated NK cells of the patient and age-matched (*n* = 2, circles) and adult (*n* = 7, triangles) HDs following 6-h culture of PBMCs with and without (not stimulated, ns) K562 cell targets. The mean ± SD is reported.

**Table 1 ijms-21-08604-t001:** Comparison of clinical features among the two siblings and previously-published cases with at least one mutation involving the same *SMARCAL1* residue (p.Arg561).

Clinical and Genetic Features		P1 (6 Years)	P2 (18 Months)	Basiratnia 2011 (8 Years)	Yue 2010 (8 Years)	Bökenkamp 2005 (12 Years)
	*SMARCAL1* genotype	*p.Arg561His/* *p.Arg561His*	*p.Arg561His/* *p.Arg561His*	*p.Arg561His/* *p.Arg561His*	*p.Arg561His/* *p.Met1Ile*	*p.Arg561Cys/* *p.Arg561Cys*
Clinical features						
Growth	IUGR	+	+	NR	+	−
	Short stature	+	+	+	+	+
Skeletal features	Short neck	+	+	+	+	+
	Short trunk	+	+	NR	+	+
	Thoracic scoliosis	+	NR	NR		+ (kyphosis)
	Lumbar lordosis	−	−	NR	NR	+
	Dysmorphic vertebrae	+	−	+	+	+
	Hypoplastic pelvis	−	−	NR	NR	NR
	Abnormal femoral heads	+	−	+	NR	+
	Microcephaly	+	+	NR	NR	NR
	Osteopenia	−	−	+	+	NR
Renal disease and malformations	Proteinuria or nephropathy	+	+	+	+	+
	FSGS	−	−	NR	+	+
	Ectopic kidney	+	−	NR	NR	NR
	Multicystic kidney	−	+	NR	NR	NR
Heart defects	Atrial septal defects	+	+	NR	NR	NR
	Pulmonary valve stenosis	−	+	NR	NR	NR
Cerebral anomalies	Ventriculomegaly	+	−	NR	NR	NR
	Hypoplasia corpus callosum	+	−	NR	NR	NR
Physical features	Broad nasal tip	−	−	NR	+	NR
	Wide and depressed nasal bridge	−	−	NR	+	+
	Protruding abdomen	−	+	+	+	+
	Pigmented macules	−	+	NR	+	NR
	Unusual hair	−	−	NR	NR	NR
	Microdontia	−	−	NR	NR	NR
	Corneal opacities	+	−	NR	NR	NR
Development	Neuro-developmental delay	+	+	NR	−	−
	Language development delay	+	−	NR	−	−
Vasculature	Headaches	+	−	NR	NR	−
	TIAs	−	−	NR	NR	−
	Strokes	+	−	NR	NR	−
Other	Hypothyroidism	−	+	+	−	−
	Non−Hodgkin lymphoma	−	−	+	NR	NR
	Recurrent infections	+	+	−	+	−
	Mesoblastic nephroma	−	+	−	NR	NR
	Extremity edema	−	−	+	+	+ (minimal)
	Hypertension	−	−	+	NR	NR

NR: Not Reported. +: feature present; −: feature absent.

**Table 2 ijms-21-08604-t002:** Analysis of WES: potentially interesting variants associated with clinical phenotype.

Genomic Position (hg19)	cDNA/ Protein Position	Gene	ROH Size (Mb)	GnomAD Frequency	PhyloP100way Vertebrate	CADD	Disease
chr2:	NM_001127207.2:c.1682G>A/ p.Arg561His	*SMARCAL1*	5.081	0.00001593	7.994	32	SIOD
217303180							
chr11:	NM_001198810.2:c.1471G>A/ p.Ala491Thr	*SLC43A1*	14.523	0.00003184	7.537	23.2	/
57254630							

**Table 3 ijms-21-08604-t003:** Immunological features of patients P1 and P2.

	P1	P2
**Sex**	Male	Male
**Age**	7 years	Died (15 months of age)
**Age at diagnosis**	5 years 6 month	1 year
**White blood cells, 10^3^/μL**	3.04	4.1
**Hemoglobin, g/L**	10.5	7.4
**Platelets, 10^3^/μL**	305	49
**Neutrophils, 10^3^/μL**	2.06	2.92
**Lymphocytes, 10^3^/μL**	0.57	0.55
	(1.2–4.7)	(3.2–12.3)
**CD3+ (PAN T), % (cells/µL)**	64.2% (0.36)	30% (0.16)
	(0.77–4.0)	(2.4–8.3)
**CD3+/α+β**	86.60%	75.40%
**CD3+/γ+σ+**	12.70%	24.6%
**CD3+CD4-CD8-, %**	1.70%	27%
**CD4, % (cells/µL)**	24.6% (0.14)	14% (0.16)
	(0.4–2.5)	(1.3–7.1)
**CD4+CD45 RA+ (naïve), %**	2.40%	17%
	(46–99)	(77–96)
**CD4+CD45 RA-CCR7+ (central memory), %**	55.60%	ND
	(0.35–100)	
**CD4+CD45 RA-CCR7- (effector memory), %**	39.40%	ND
	(0.27–18)	
**CD4+CD45 RA+CCR7- (terminal effector memory), %**	2.37%	ND
	(0.0031–1.8)	
**CD3+CD4+CD31+CD45 RA+ (recent thymic emigrant) %**	2%	ND
	(41–81)	
**CD8, % (cells/µL)**	26.6% (0.15)	10% (0.05)
	(0.2–1.7)	(0.4–4.1)
**CD8+CD45 RA+ (naïve), %**	2%	ND
	(16–100)	
**CD8+CD45 RA-CCR7+ (central memory), %**	1.57%	ND
	(1–6)	
**CD8+CD45 RA-CCR7- (effector memory), %**	80%	ND
	(5–100)	
**CD8+CD45 RA+CCR7- (terminal effector memory), %**	16.50%	ND
	(15–41)	
**CD56+16+CD3- (NK), % (cells/µL)**	15% (0.08)	26% (0.14)
	(0.012–0.34)	(0.0075–0.33)
**CD19 (PAN B), % (cells/µL)**	18.4% (0.1)	42% (0.23)
	(0.10–0.80)	(0.11–7.7)
**CD19+IgD+CD27- (B naïve)**	85%	94%
	(47.3–77.0)	(76.5–94.7)
**CD19+IgD+CD27+ (B memory)**	8.80%	1%
	(5.2–20.4)	(3.0–10.7)
**CD19+IgD-CD27+ (switched B memory)**	6.16%	1.8%
	(4.7–21.2)	(1.4–11.9)
**CD19+CD21+CD38- (B CD21+low)**	0.80%	ND
	(5.9–25.8)	
**CD19+IgM++CD38++ (B transitional)**	18.40%	30%
	(4.6–8.3)	(3.6–12.7)
**CD19+IgM-+CD38++ (B plasmablast)**	0.10%	1%
	(0.6–5.3)	(0.4–5.5)
**IgM**	0.10 g/L	0.56 g/L
	(0.03-0.20)	(0.02–0.18)
**IgA**	0.10 g/L	0.10 g/L
	(0.02–0.20)	(0.02–0.15)
**IgG**	0.64 g/L *	1.16 g/L *
	(0.52–1.49)	(0.42-1-1)

FISH, fluorescence in situ hybridization; ND, not done; PHA, phyto hemagglutinin; TCR, T-cell receptor; WBC, white blood cell. Normal values for serum immunoglobulin concentrations derive from [22], T cell subsets from [23], and B cell subsets from [24]. * Under monthly IVIG infusion.

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
