# Peer review of "Expanding Phenotype of Schimke Immuno-Osseous Dysplasia: Congenital Anomalies of the Kidneys and of the Urinary Tract and Alteration of NK Cells"

_ijms, 2020, doi:10.3390/ijms21228604_

Round 1
Reviewer 1 Report
The manuscript ID: ijms_959289 – titled: “EXPANDING PHENOTYPE OF SCHIMKE IMMUNO OSSEOUS DYSPLASIA: CONGENITAL MESOBLASTIC NEPHROMA AND ALTERATION OF NK CELLS” reports two siblings with Schimke immuno-osseous dysplasia (SIOD) carrying the same homozygous missense variant in SMARCAL1 associated with different substantial renal and immunological phenotypic differences along with features not previously associated with SIOD.
The authors performed an extensive characterization of the immunophenotype showing combined immunodeficiency characterized by a profound lymphopenia, lack of thymic output, defective IL-7R expression, disturbed B plasma cells differentiation and immunoglobulins production in addition to an altered NK-cell phenotype and function.
Even though an attempt of genotype/phenotype correlation was previously supposed for SIOD, suggesting milder effects for missense variants and severe consequences for LOF molecular events, the results reported here support other observation of intra and interfamilial clinical variability showing that disease severity or outcome cannot be easily predicted by SMARCAL1 genotype.
Considering that our knowledge about SIOD is still limited and considering the rarity of the disease, the results reported here contribute to extend the phenotypic spectrum of features associated to SMARCAL1 mutations and to better characterize the underlying immunologic disorder with critical implications for therapeutic and management strategies.
The manuscript is well written and well detailed. The English language is appropriate and understandable. One point has not been well addressed in the manuscript and for this reason, I suggest the publication after a major revision.
Major revision
In the Discussion section, considering the unusual phenotypical features in the two siblings, the authors should consider and critically discuss the possibility of a genetic dual diagnosis rather than claiming for an expansion of the phenotype, in particular for the congenital mesoblastic nephroma.
Although no homozygous variants in disease-genes, except the SMARCAL1 variant, were found known or predicted to be pathogenic or previously associated to congenital kidney defects, the mesoblastic nephroma found in P2 could be an independent event due to another variant in another gene not present in P1. Unfortunately, WES has not been carried out in P2 and the reason for this should also be discussed in the WES section.
The authors focused their search on other homozygous variants that could be implicated in the congenital kidney and heart defects phenotype, but they should also report about putative causative heterozygous variants, assuming a dominant or recessive (compound heterozygous) inheritance pattern.
In this context, it would be more appropriate to change the title, and instead of “congenital mesoblastic nephroma”, put “Congenital Anomalies of the Kidneys and of the Urinary Tract”.
Minor revisions
In Table 1, please use the italic for the genes’ name and specify the hg used for the genomic position.
In Figure 3, the biallelic variant in P1 is not correctly reported.
In Figure 3 legend and the Abstract, please report the variant nomenclature consistently with the figure and all the text, i.e. p. Arg561His instead of p.R561H.
In WES findings section, lines 157-158, the references 21 is 22 and 22 is 29. So, the authors should accurately revise the References both in the text and in the list.
In Table 2, please add a – sign or NA for the missing features so that these are consistently reported within the table.
Please specify the acronym HD the first time used in the text.
In the Discussion section, please specify better what is CAKUT, since in the Abstract the acronym was not reported.
In the figures’ legends, please pay attention to the indication of the different panels, i.e. A, A) or (A), so that they are consistent throughout the manuscript.
Author Response
Thank you for your kind and useful comments and revisions.
Please find attached our original revised manuscript entitled “Expanding phenotype of Schimke immuno-osseous dysplasia: Congenital Anomalies of the Kidneys and of the Urinary Tract and alteration of NK cells “ (title changed).
Line 287-291: answering to REV1 major revisions, we add in the text the following sentences “The evaluation of heterozygous variants excluded the co-occurrence of dominant or recessive (through compound heterozygosity) genetic diseases modifying the typical SIOD phenotype. Unfortunately, the impossibility to extend WES analysis to P2 due to technical issues, prevented us to investigate additional genetic variants underlying his atypical manifestations, especially the congenital mesoblastic nephroma.”
Line 379-383: answering to REV1 major revisions, we add in the text the following sentences “The same filtering parameters (frequency, conservation and functional prediction scores) were used to identify candidate heterozygous variants potentially contributing to atypical clinical manifestations through a dominant or a recessive (compound heterozygosity) mechanism. The quality and the amount of the available DNA sample of P2 were not adequate for WES analysis.”
Table 2 with WES analysis, Hg used for genomic position was specified
In Figure 3, the biallelic variant nomenclature in P1 has been corrected.
In Figure 3 legend and the Abstract, the variant nomenclature has been corrected with p. Arg561His
References and bibliography has been corrected and updated based on reviewer comments
In Table 2, a – sign or NR has been added for the missing features
Line 267: acronym CAKUT is for Congenital Anomalies of the Kidneys and of the Urinary Tract
Figures’ legends, has been modified and homogeneized regarding the indication of the different panels, i.e. (A), so now that they are consistent throughout the manuscript.

Reviewer 2 Report
The article by Cristina Bertulli et al focuses on Schimke immuno-osseous dysplasia which is a rare disorder linked to biallelic variants in SMARCAL1. The study describe the case of two siblings born from consanguineous parents and highlights features not previously associated with Schimke immuno-osseous dysplasia such as Congenital Anomalies of the Kidneys and of the Urinary Tract and congenital mesoblastic nephroma.
Overall, the manuscript is well written and presents an interesting and extensive characterization of the immunophenotype of the two brothers.
However, I have a few questions/comments.
* Line 106-109: “.The clinical features of the patient associated with the typical skeletal abnormalities (left convex thoracic scoliosis, abnormal femoral heads and dismorphic vertebrae), led to the suspicion of Schimke-Immuno-Osseous Dysplasia (SIOD) (Table 1).”
I don’t understand why the authors refer to Table 1 whereas the clinical features are listed in Table 2. If, in fact they refer to Table 2, I suggest to reverse the order of Table 1 and 2 since Table 2 would be mentioned on line 109 and Table 1 on line 147.
* Line 207: “§ The normal value indicated was measured in the same experiment when two controls were used for allogeneic stimulation”
I don’t see the symbol § in Table 3.
* Line 227-228: “… to measure the frequency of of NK-cell ….” should be replaced by “… to measure the frequency of NK-cell ….”
* Line 274-275: “Clinical features of SIOD patients with variants involving the Arg561 residue are summarized in Table 1
As for the comment related to line 106-109 above, I don’t understand why the authors refer to Table 1 whereas the clinical features are listed in Table 2.
* Line 427 to 434: I suggest to replace “partecipated” by “participated” and “e” by “and”.
Tables
* Table 2 – line 159
I suggest to correct the SMARCAL 1 genotype of P1: p.Arg561Hi/ should be replaced by p.Arg561His/
* Table 3 – within the 10 first lines
I suggest using the proper notation for the number in power: 103 instead of 10^3
References
The number of the references are given twice !!!
I suggest to homogenize the way the initials of the authors are written throughout the references section. For instance, the authors of reference 1 and reference 2 are not written the same way since for reference 1 the initials are placed after the last name and for reference 2 the initials are placed before the last name. Likewise, some of the references give the month of the publication while others don’t, and articles pages are not always given the same way.
* reference 6 → use of the first name and not the initials.
* reference 8 → correct the article pages
S29-S27 instead of 29S-37S.
* reference 9 → correct the journal reference
Sep 13;9:27 instead of Sep 13;9(1):27
* reference 15 → correct the journal reference (if you choose to include the month of publication ....)
Nat Genet. 2002 Jan;30(2):215-220 instead of Nat Genet. 2002 Feb;30(2):215–20
* reference 24 → some initials of the authors are missing (Schatorje, Gemen & Driessen)
E J H Schatorjé, E F A Gemen, G J A Driessen, J Leuvenink, R W N M van Hout, M van der Burg, E de Vries
* reference 26 → correct the initial of one of the authors
Finocchi A and not F
* reference 30 → check the issue of the journal (I think there is a mix with the doi)
From pubmed, I found: Pediatr Blood Cancer. 2017 Jul;64(7).
* reference 31 → check the issue of the journal (I think there is a mix with the doi)
From pubmed, I found: Pediatr Blood Cancer. 2018 Apr;65(4).
* reference 34 →Why are some names underlined ?
Author Response
Thank you for your kind and useful comments and revisions.
Please find attached our original revised manuscript entitled “Expanding phenotype of Schimke immuno-osseous dysplasia: Congenital Anomalies of the Kidneys and of the Urinary Tract and alteration of NK cells “ (title changed).
Line 110: table 1 with clinical features of P1, P2 and other SIOD patients
Line 164: table 2 switched to table 1
Line 210: § in Table 3 eliminated, mistake
Line 231-232: “… to measure the frequency of of NK-cell ….” has been replaced by “… to measure the frequency of NK-cell ….”
Line 443-450: we replace “partecipated” by “participated” and “e” by “and”.
Line 110: we correct the SMARCAL 1 genotype of P1: p.Arg561Hi/ has been replaced by p.Arg561His/
Line 205: table 3. Edited the proper notation for the number in power (103instead of 10^3)
References and Bibliography has been updated based on reviewer comments

Round 2
Reviewer 1 Report
I revised the new version of the manuscript and the authors addressed all my concerns. Therefore I think the manuscript now warrants publication in IJMS in present form.
Best regards